



# Improving Wind Speed Availability of a Six-Beam Doppler Lidar

Mohammadreza Manami[1,2], Guillaume Léa[2], Jakob Mann[1], Mikael Sjöholm[1], and Guillaume Gorju[2]

[1]DTU Wind and Energy Systems, Technical University of Denmark, Roskilde, Denmark
[2]Lidar Division, Lumibird SA, Lannion, France

**Correspondence:** Mohammadreza Manami (manami@dtu.dk)

**Abstract.** A simple adaptive variant of the Doppler Beam Swinging (DBS) method is presented to enhance the availability of wind velocity measurements in profiling lidars. The adaptive method dynamically selects beams with sufficient signal-to-noise ratios (SNR) for wind velocity reconstruction, instead of the standard approach, which discards a complete scan when one beam falls below the SNR threshold. The adaptive method was validated in two measurement campaigns at the Østerild wind turbine test field in Denmark using three BEAM 6x profiling lidars from Lumibird. In the first campaign, a lidar measured up to 500 m in proximity to a meteorological mast; in the second campaign, the first lidar was replaced by two other lidar units to increase the maximum measurement range up to 1 km. Validation against cup anemometers and wind vanes at four different heights of the met mast showed excellent agreement for mean wind speed and wind direction, with results similar to those from the standard approach. Availability assessments indicated improvements for all three lidars at high altitudes, showing a maximum increment of 16.9 percentage points over the standard approach. Due to its simplicity, the adaptive method can be implemented in lidar software without requiring any hardware modifications.

## 1 Introduction

As modern wind turbines are increasingly deployed at heights spanning the entire atmospheric boundary layer (ABL), understanding the physics of the entire ABL has become more essential (van Kuik et al., 2016; Veers et al., 2019, 2023). In recent years, wind Doppler lidars have emerged as a popular and cost-effective alternative to meteorological masts, offering the capability to measure wind profiles at significantly greater heights. However, one of the challenges in extended ranges of profiling lidars is the weakening of atmospheric backscatter signals, due to beam divergence and aerosol-related attenuation of the laser beam (Ceolato and Berg, 2021; Measures, 1984). The aim of this study is to introduce a straightforward algorithm that enhances the availability of reconstructed wind velocities by adaptive selection of the maximum beam numbers with adequate SNR, without requiring any hardware modifications.

Doppler Beam Swinging (DBS) is a well-established and widely used method for reconstructing wind velocities from Doppler velocity measurements (Lehmann and Brown, 2021; van Dooren, 2022). DBS requires at least three Doppler velocities from non-coplanar beams to reconstruct the three components of the wind vector. Most commercial pulsed profiling lidars use more than three beams to improve the accuracy of this reconstruction. In this study, we propose an adaptive algorithm for selecting among the six beams of the Lumibird profiling lidar, aimed at improving the availability of reconstructed wind velocities, particularly at higher altitudes. The current standard DBS setup on this instrument rejects an entire scan if





any of the Doppler velocities from inclined beams falls below the factory-set SNR threshold. This becomes a limitation at higher altitudes, where not all beams consistently meet the SNR requirement, resulting in a loss of usable Doppler velocities. Our adaptive approach mitigates this issue by selectively using only those beams that satisfy the SNR requirement, thereby
maintaining reliable wind velocity retrievals while increasing data availability.

## 2  Methodology

The Doppler Beam Swinging (DBS) technique in pulsed lidar is conceptually adopted from the velocity azimuth display (VAD) method, which was originally developed for use in radar systems (Lhermitte, 1961; Browning and Wexler, 1968). Both the VAD and DBS techniques involve measuring mean Doppler velocities at positions that form a cone above the instrument
which is placed in the apex. In the VAD technique, the lidar beam rotates continuously at a fixed elevation angle, while the DBS approach typically relies on a discrete set of static beams. Under the assumption of a horizontally homogeneous and steady wind field at a given height, the complete wind vector can be retrieved by minimizing the squared differences between the measured radial velocities and the corresponding projections of the wind vector onto the unit vectors aligned with each beam.

$$\chi^2 = \sum_{i=1}^{N} \frac{(\boldsymbol{u} \cdot \boldsymbol{r}_i - u_{ri})^2}{\sigma_{ri}^2}, \tag{1}$$

In the above equation, $\boldsymbol{u} = (u, v, w)$ denotes the mean wind velocity vector to be determined, $N$ is the number of lidar beams, $u_{ri}$ is the radial velocity measured along the $i$-th beam, and $\sigma_{ri}$ represents its associated uncertainty (Newsom et al., 2017). The unit vector $\boldsymbol{r}_i$, which defines the orientation of each lidar beam, is given by:

$$\boldsymbol{r}_i = (\cos\alpha_i \cos\theta_i, \cos\alpha_i \sin\theta_i, \sin\alpha_i), \tag{2}$$

where $\alpha_i$ and $\theta_i$ are the elevation and azimuth angles, respectively. The BEAM 6x Lumibird lidar includes five beams with
elevation angles near 60 degrees, distributed evenly in azimuth with 72-degree spacing on the full 360-degree circle (see Figure 2). It also features a single vertical beam with a 90-degree elevation. The beams are numbered from 1 to 6, where beams 1 through 5 represent the inclined beams arranged in increasing azimuth angles starting from zero degrees, and beam 6 corresponds to the vertical beam.

In the current standard DBS setup on the BEAM 6x, valid measurements of all individual radial velocities from inclined
beams are required to perform DBS; otherwise, the entire scan is rejected. This study propose a simple method that applies DBS using the maximum available number of beams, which is particularly useful in scenarios where certain beams fail to reach the SNR threshold.

The procedure of the proposed adaptive DBS is presented in Figure 2. The algorithm begins by applying a sliding window across six consecutive Doppler velocity measurements from six beams at a specific height. First, it filters out any radial ve-
locities with an SNR below a predefined factory threshold. If at least three valid Doppler velocities remain, a DBS process is performed. Before applying the DBS method, a quality check is adopted to verify the reliability of the estimated wind components. This step is designed to identify so-called ill-conditioned scenarios, which occur when only three beams are available



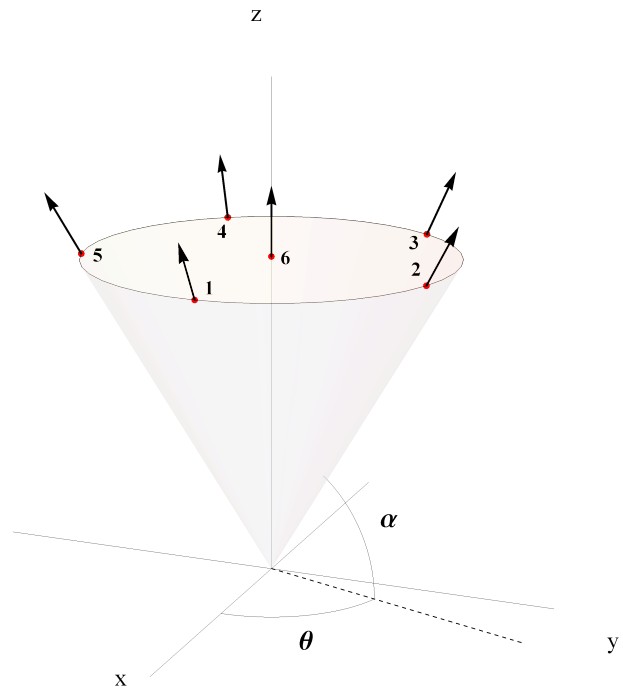

**Figure 1.** Beam directions for BEAM 6x Lumibird lidar. The instrument has five inclined beams at approximately 60° elevation, evenly spaced every 72° in azimuth from 0° to 360° (numbers 1 to 5, respectively), along with a vertical beam at 90° elevation (number 6).

and their azimuthal orientations are poorly distributed, typically when the line-of-sight vectors lie nearly in one plane. This algorithm effectively excludes only five sets of three-beam, which correspond to the least accurate cases among the 42 possible

combinations (not shown here). The five excluded combinations are 1-3-6, 1-4-6, 2-4-6, 2-5-6, and 3-5-6 (see Figure 1), as they are considered to be ill-conditioned scenarios. This algorithm continues until either a valid solution is found or fewer than three beams remain, at which point a solution is no longer feasible.

Another criterion that can be discussed is the root mean square error $RMSE$ of the fit to Doppler velocities, which can be calculated after performing DBS. We do not recommend relying on this metric, as high turbulence can also result in a high

$RMSE$, making it difficult to differentiate between a poor fit and turbulence.

Availability $A$ is defined as the ratio of filtered instantaneous reconstructed velocities $N_f$ to the total number of complete scans $N_t$. For the adaptive method, $N_f$ corresponds to the number of reconstructed wind velocities remaining after applying all the filters shown in Figure 2. In the standard method, $N_f$ refers to the number of reconstructed wind velocities from the five inclined beams after the intensity filter is applied.

$$A = \frac{N_f}{N_t} \times 100 \tag{3}$$



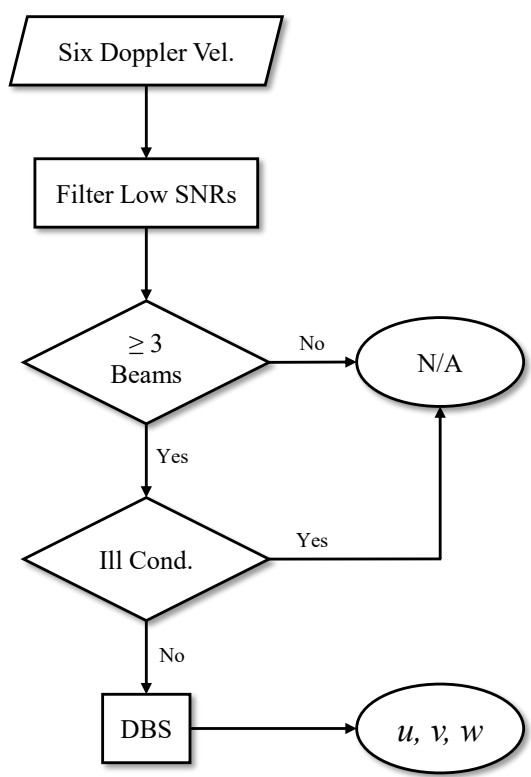

**Figure 2.** Algorithm of adaptive DBS.

The proposed adaptive DBS approach was evaluated through field experiments using three BEAM 6x profiling lidars during two measurement campaigns at the Østerild wind turbine test field (Peña, 2019). Ten-minute mean wind speeds and wind directions obtained from the lidars with adaptive DBS were compared against reference measurements from cup anemometers and wind vanes installed on a nearby meteorological mast. Also, results from the standard DBS method are included. The following filters were applied to both standard DBS and adaptive methods to ensure the validity of the comparison with reference instruments, which is also aligned with the DTU calibration report (Hansen and Yankova, 2024).

– Lidar Data Availability: Only 10-minute intervals containing at least 75% of the total possible complete scans were included in the analysis to ensure the reliability and representativeness of the mean wind speeds and wind directions.

– Wake-Free Sector: Wind directions from the westerly sector (195° to 355°) were selected to avoid interference from a nearby wind turbine and meteorological mast. This sector is broader than the range of 240° to 300° recommended in the DTU calibration report, in order to provide more 10-minute average data points for the validation of adaptive DBS.





- Wind Speed Range: The analysis was limited to 10-minute average wind speeds between 4 m/s and 16 m/s, consistent with the valid operating range of standard cup anemometers as defined by the IEC standard (IEC, 2022).

The DTU calibration report (Hansen and Yankova, 2024) implements an additional filtering criterion to remove the effects of potential icing, specifically excluding wind speeds collected at temperatures below 2°C. This temperature-based filter was not applied in our analysis, as it only affected a small portion of the measurements. To validate and certify the estimated mean wind speeds and wind directions, the linear regression slope and the coefficient of determination (denoted $R^2$), between the lidar measurements and the reference instruments should typically fall within the acceptable range of 0.98 to 1.02. In this work, the proposed method was validated at four heights ranging from 40 m to 244 m, after which data availability was examined for heights up to 980 m.

## 3 Field Experiment

The field experiment was conducted at the Danish National Test Station for Large Wind Turbines in Østerild, Northern Jutland, Denmark. Three BEAM 6x Wind Sciences (HALO Photonics by Lumibird, 2024) profiling lidars were installed west of the Light Mast North over two separate deployment periods. The profiling lidars are hereafter identified by the last three digits of their serial numbers: 205, 320, and 325. The first campaign, using BEAM 6x-205, took place from October 13, 2023, to January 9, 2024, during which wind profiles were measured up to 500 m. Subsequently, lidar units 320 and 325 were installed in the same place, from October 3 and November 13, respectively, extending the measurement range up to 1 km until December 6, 2024. This extension, however, is out of the designed range of 500 m for the BEAM 6x system. Figure 4 displays the installation site of BEAM 6x-205, where the two other units were later installed at approximately the same position.

The BEAM 6x lidar system operates with a pulse width of 150 ns, a laser power of 20 μJ, and a pulse repetition frequency of 10 kHz, while its acquisition board samples data at 50 MHz. The orientation of six beams is illustrated in Figure 1. This system performs a complete scan every 5.5 seconds, providing a high-resolution wind profile at every 2.6 m range gates in altitude. In the field experiment, Doppler velocities are collected from three lidars to implement adaptive DBS and to compare its availability with the standard DBS method.



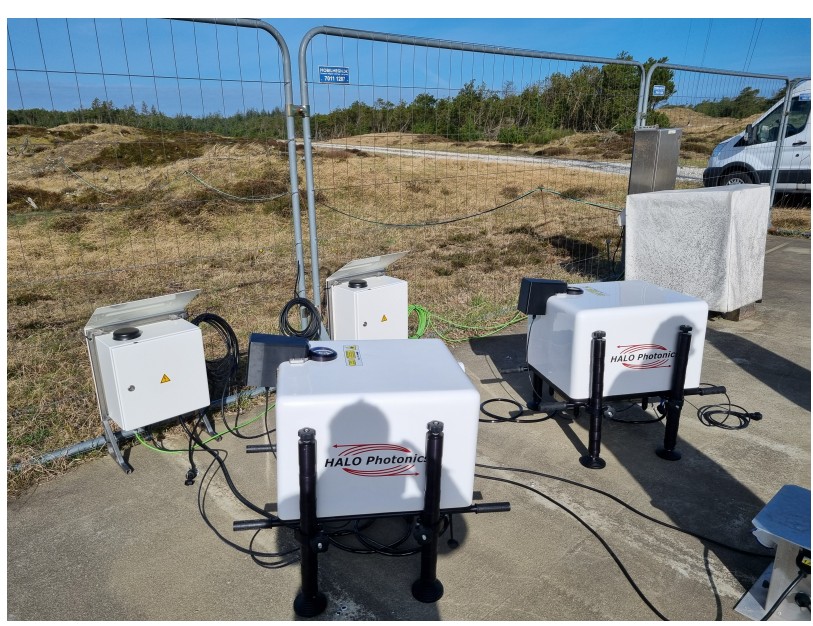

**Figure 3.** Two BEAM 6x profiling lidars manufactured by Lumibird with serial numbers of 320 and 325, installed at the Østerild test site.

**Figure 4.** Satellite image of the experimental field at Østerild test site (Denmark) from Bing Maps, copyright 2025 © Microsoft. The BEAM 6x profiling lidar (white dot) with serial number 205 is located near the Light Mast North (green dot). Two other units were later installed at approximately the same location.





## 4    Results and Discussion


The adaptive DBS method was validated by comparing 10-minute averaged wind speeds and directions against reference measurements from cup anemometers and wind vanes, following the filtering procedure described in Section 2. Validation results are summarized in Tables 1, A1, and A2 for BEAM 6x-205, 320, and 325, respectively. For comparison, the results from the standard DBS, similar to the current implementation in BEAM 6x instruments, are also included.

The evaluation metrics consist of the slope of the least-orthogonal-squares fit, $R^2$, and $RMSE$, computed at four heights: 40 m, 106 m, 178 m, and 244 m. These heights were selected because both reference cup anemometers and wind vanes are available at these levels. The adaptive method demonstrates excellent agreement with the reference instruments for both wind speed and wind direction, and its performance closely matches that of the standard DBS method. At lower heights, data availability is similar for the two methods, but as height increases, the adaptive method provides greater availability. For

instance, at 244 m in BEAM 6x-320, availability increases from 78.7% with the standard approach to 88.8% with the adaptive approach (Table A1).

Across all reference heights and among the three units, only BEAM 6x-320 at 40 m exhibited a slight degradation in wind direction RMSE, which is caused by a small number of outliers in the regression analysis. An additional enhancement to the adaptive DBS could involve implementing a spike filter that compares the Doppler velocity of each beam with its two

temporal neighbors. The spike filter could help suppress weak velocity estimates while preserving a similar high availability. However, this filter was not included in the present analysis since this study is focused on the adaptive beam selection algorithm and the combined effects of spike filtering and SNR thresholds is a big topic for investigation in itself. Also, spike detection for each beam requires access to the radial velocity measurement from the subsequent time step, which becomes available approximately five seconds later in BEAM 6x, once a scan is completed. As a result, implementing such analgorithm would

introduce a slight delay of a few seconds relative to real-time measurements. In the adaptive DBS algorithm, the objective was to apply only minor modifications to the original BEAM 6x algorithm, especially without altering the factory-defined SNR threshold. While small adjustments to the SNR threshold could further improve availability, such changes would risk compromising velocity estimation accuracy. The original design specifications of the BEAM 6x lidar target wind profiling up to 500 m; therefore, some of the design parameters might need reconsideration when aiming for higher altitudes.

As an example of regression analysis, Figure 5 presents the 10-minute averaged wind speed and wind direction at a height of 244 m for BEAM 6x-205 against reference instruments, in order to compare results from the standard and adaptive DBS methods. The distribution of data points, regression lines, and derived statistical metrics is comparable between the two approaches, supporting the validity of the proposed adaptive DBS technique. As noted in Section 2, the DTU calibration report limits the analysis sector to 240°–300°. However, the present study considers a broader sector to increase the number of available 10-

minute data points. This broader inclusion may contain distortions from the Light Mast North guy wires located at 210° and 330°. In the wind direction regression plots, localized scatter appears near these angles, likely due to the distortion effect of the guy wires.



**Table 1.** Validation of DBS methods using Doppler velocities from BEAM 6x-205 against cup anemometer and wind vane measurements.

| height [m] | method | $A$ [%] | variable | slope [-] | $R^2$ [-] | RMSE [m/s, °] |
|---|---|---|---|---|---|---|
| 40 | Standard | 99.9 | WS | 1.028 | 0.989 | 0.32 |
| | | | WD | 1.003 | 0.999 | 1.34 |
| | Adaptive | 100.0 | WS | 1.028 | 0.989 | 0.32 |
| | | | WD | 1.003 | 0.999 | 1.33 |
| 106 | Standard | 99.8 | WS | 1.000 | 0.997 | 0.16 |
| | | | WD | 1.001 | 0.999 | 0.98 |
| | Adaptive | 99.9 | WS | 1.000 | 0.997 | 0.16 |
| | | | WD | 1.001 | 0.999 | 0.98 |
| 178 | Standard | 98.0 | WS | 0.994 | 0.996 | 0.19 |
| | | | WD | 0.999 | 0.999 | 0.95 |
| | Adaptive | 98.8 | WS | 0.994 | 0.996 | 0.19 |
| | | | WD | 0.999 | 0.999 | 0.98 |
| 244 | Standard | 92.5 | WS | 0.995 | 0.997 | 0.18 |
| | | | WD | 0.996 | 0.998 | 1.43 |
| | Adaptive | 95.5 | WS | 0.995 | 0.997 | 0.19 |
| | | | WD | 0.996 | 0.998 | 1.50 |






**Figure 5.** Ten-minute mean wind speeds (5a) and wind directions (5b) from adaptive DBS applied to Doppler velocities from BEAM 6x-205 profiling lidar, compared with reference measurements from cup anemometers and wind vanes mounted at 244 m on the Light Mast North. Corresponding plots for standard DBS are shown for mean wind speeds (5c) and wind directions (5d).





Following the validation of the adaptive DBS, availability profiles were computed for all three lidar units. Figure 6a shows the availability of BEAM 6x-205 across ranges from 40 m to 500 m, reported in 10 m increments. For this unit, the availability

obtained with the standard and adaptive DBS methods is nearly identical up to approximately 150 m. Beyond this range, the adaptive method provides a consistent improvement, with the difference gradually increasing and reaching a maximum of 15.3 percentage points at 500 m height.

A similar analysis was performed for BEAM 6x-320 (Figure 6b) and 325 (Figure 6c), with availability assessed in 20 m range gates up to 980 m altitude. The results show that the difference in wind velocity availability between the two methods

starts from lower altitudes and increases with height. For BEAM 6x-320, the maximum improvements in availability with the adaptive method are observed to be 16.9 percentage points. Similarly, BEAM 6x-325 exhibited maximum improvements of 12.9%. The relative availability differences between two methods are also presented in Figure 6d. As SNR decreases with altitude, the adaptive DBS delivers higher availability, reflecting the benefit of dynamically selecting beam combinations rather than relying exclusively on beams 1 to 5.

Across all three lidars, the adaptive approach consistently achieves availability equal to or greater than that of the standard method. Differences in availability profiles between instruments may be partly attributed to atmospheric variability, since measurements were collected during different time periods. BEAM 6x-320 and 325 show greater similarity, likely due to a one-month overlap in their campaigns. Instrument-specific hardware and optical performance characteristics may also contribute to the observed variability among devices.





**Figure 6.** Availability profiles obtained using both adaptive and standard DBS applied to Doppler velocities from BEAM 6x-205 (6a), BEAM 6x-320 (6b), and BEAM 6x-325 (6c). Relative differences between the two methods across the three instruments at various altitudes are also shown (6d).





A distribution analysis of the number of beams used for adaptive DBS applied to Doppler velocities of BEAM 6x-320 (Figure 7) highlights the importance of considering three-, four-, and five-beam combinations. Standard DBS availability is indicated by a black arc, representing the fixed configuration of beams 1–5, similar to implementation in the BEAM 6x software. Interestingly, for the first two heights, after the full six-beam combination, the five-beam combinations, including the standard 1–5 set as well as other five-beam arrangements, account for a higher proportion of availability than the other two

groups. At 980 m, the contributions from the three-beam and five-beam combinations are equal, and both are slightly lower than those from the four-beam arrangements. This ranking may change depending on atmospheric conditions and the SNR of each beam.

In this study, we applied a conservative algorithm that discards three-beam combinations with unit vectors under ill con-ditions. Nevertheless, the difference between the filtered algorithm and the unfiltered version is negligible in terms of both

accuracy and availability, as the proportion of excluded configurations is very low. Simple combinations of ill-conditioned beams (e.g., a fixed combination of beams 1, 4, and 6) yield the lowest accuracy among all 42 configurations (not shown). Therefore, if the proportion of such ill-conditioned combinations increases in the adaptive method, the overall accuracy may be slightly reduced.

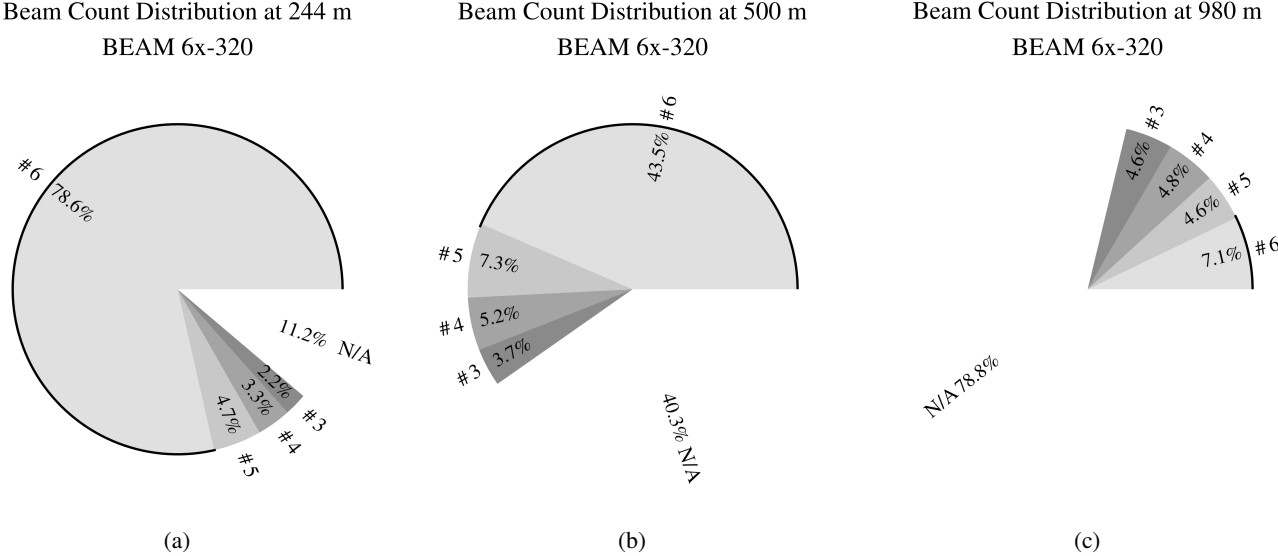

**Figure 7.** Distribution of beam counts used in adaptive DBS applied to Doppler velocities from BEAM 6x-320 at 244 m (7a), 500 m (7b), and 980 m (7c). The availability of the fixed beam configuration 1–5 is indicated by black arcs, consistent with the implementation in the BEAM 6x software.





## 5    Conclusion

In this study, a simple method was introduced to enhance the availability of reconstructed wind velocities in pulsed profiling lidars. The approach is an adaptive variant of DBS, which uses only beams meeting the SNR requirement for wind velocity reconstruction. In contrast to the standard DBS method, which discards an entire scan if a single inclined beam falls below the SNR threshold, the adaptive approach dynamically selects combinations of three to six beams. This maximizes the use of Doppler velocity measurements and increases overall data availability.

The method was validated using three BEAM 6x profiling lidars from Lumibird at the Danish National Test Station for Large Wind Turbines in Østerild, Denmark. The first lidar unit was deployed for nearly three months close to a meteorological mast. After this initial campaign, two additional units were installed at the same site to extend measurements reaching up to 1 km. One unit collected data for two months, the other for one month, with a one-month overlap. Ten-minute averages of wind speed and direction from all three lidars were compared with cup anemometers and wind vanes at four heights on the mast,

demonstrating excellent agreement with the reference instruments.

Following this validation, the availability of reconstructed instantaneous wind velocities was compared between the standard and adaptive approaches. The adaptive approach consistently improved availability for all three lidars at higher altitudes. Assessing the availability of two units up to the range gate of 980 m showed that the maximum differences between the two methods were greater than 16.9 percentage points.

Analysis of the beam combinations utilized in the adaptive method for BEAM 6x-320 showed that six-beam combinations dominate, followed by five-, four-, and three-beam combinations. At 980 m, the three- and five-beam contributions are equal and slightly below the four-beam contribution. At this height, the three- and four-beam contributions are of the same order of magnitude as those from the five- and six-beam arrangements, highlighting their role in enhancing availability. Although the validation process revealed no significant difference when removing ill-conditioned three-beam combinations, due to their low

occurrence in the algorithm, we opted to exclude them. This precaution ensures that, should their proportion increase in future datasets, accuracy will not be compromised.

Future research could explore these ill-conditioned combinations in more detail to improve understanding of occasional weak estimates. A detailed analysis of the SNR threshold and the necessity of spike filtering is also recommended for future investigations. Moreover, it would be valuable to determine the minimum number of beams required in a profiling lidar to

achieve comparable accuracy and data availability.



# Appendix A

**Table A1.** Validation of DBS methods using Doppler velocities from BEAM 6x-320 against cup anemometer and wind vane measurements.

| height [m] | method | $A$ [%] | variable | slope [-] | $R^2$ [-] | RMSE [m/s, °] |
|---|---|---|---|---|---|---|
| 40 | Standard | 97.1 | WS | 1.034 | 0.979 | 0.36 |
| | | | WD | 0.996 | 0.998 | 1.80 |
| | Adaptive | 98.9 | WS | 1.035 | 0.977 | 0.37 |
| | | | WD | 0.996 | 0.997 | 2.18 |
| 106 | Standard | 96.5 | WS | 1.003 | 0.997 | 0.16 |
| | | | WD | 0.994 | 0.997 | 1.90 |
| | Adaptive | 99.2 | WS | 1.003 | 0.997 | 0.16 |
| | | | WD | 0.994 | 0.997 | 1.90 |
| 178 | Standard | 88.3 | WS | 0.997 | 0.998 | 0.15 |
| | | | WD | 0.991 | 0.995 | 2.57 |
| | Adaptive | 94.8 | WS | 0.997 | 0.998 | 0.16 |
| | | | WD | 0.991 | 0.995 | 2.60 |
| 244 | Standard | 78.7 | WS | 0.995 | 0.997 | 0.18 |
| | | | WD | 0.989 | 0.993 | 3.16 |
| | Adaptive | 88.8 | WS | 0.995 | 0.997 | 0.19 |
| | | | WD | 0.989 | 0.993 | 3.20 |





**Table A2.** Validation of DBS methods using Doppler velocities from BEAM 6x-325 against cup anemometer and wind vane measurements.

| height [m] | method | $A$ [%] | variable | slope [-] | $R^2$ [-] | RMSE [m/s, °] |
|---|---|---|---|---|---|---|
| 40 | Standard | 99.7 | WS | 1.012 | 0.987 | 0.27 |
| | | | WD | 0.990 | 0.994 | 3.11 |
| | Adaptive | 100.0 | WS | 1.012 | 0.987 | 0.28 |
| | | | WD | 0.990 | 0.994 | 3.11 |
| 106 | Standard | 97.1 | WS | 1.004 | 0.997 | 0.16 |
| | | | WD | 0.988 | 0.994 | 3.31 |
| | Adaptive | 98.5 | WS | 1.004 | 0.997 | 0.16 |
| | | | WD | 0.988 | 0.994 | 3.31 |
| 178 | Standard | 88.7 | WS | 0.995 | 0.996 | 0.18 |
| | | | WD | 0.986 | 0.991 | 3.81 |
| | Adaptive | 92.5 | WS | 0.995 | 0.996 | 0.18 |
| | | | WD | 0.986 | 0.992 | 3.82 |
| 244 | Standard | 80.5 | WS | 0.995 | 0.996 | 0.20 |
| | | | WD | 0.984 | 0.988 | 4.45 |
| | Adaptive | 86.0 | WS | 0.995 | 0.996 | 0.20 |
| | | | WD | 0.984 | 0.988 | 4.45 |





*Data availability.* Data will be provided upon request, subject to the decision of the Technical University of Denmark and Lumibird SA.

*Author contributions.* MM: draft, methodology, software, and analysis. GL and JM: conceptualization, methodology, supervision, and funding acquisition. MS: methodology, supervision, analysis, review, and funding acquisition. GG: supervision, review, and funding acquisition.

*Competing interests.* JM holds a chief editor position in the Wind Energy Science (WES) journal. MM, GL, and GG are employed by Lumibird SA.

*Acknowledgements.* This project has received funding from the European Union's Horizon Europe research and innovation program under the Marie Skłodowska-Curie grant agreement No 101119550. The authors acknowledge Jesper Grossmann Hansen, Hazal Ozcan, Ginka Georgieva Yankova, Valur Aðalsteinsson Vestmann, Allan Djernes Blaabjerg, and other contributors from the Testing and Calibration (TAC) section of DTU Wind and Energy Systems for performing field experiments and collecting the required datasets. Contributions from Poul Hummelshøj and Hans-Juergen Kirtzel (Metek Nordic Aps) are also acknowledged.





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
