# Peer review of "Improving Wind Speed Availability of a Six-Beam Doppler Lidar"

_Wind Energy Science, 2025_

## Author Comment (AC1)

**Author Comments:**

We would like to thank the referees for their valuable feedback, which has enhanced the clarity of this manuscript. We have carefully considered each comment and revised the manuscript accordingly. Our point-by-point responses are provided below.

**RC1: https://doi.org/10.5194/wes-2025-165-RC1, 03 Oct 2025**

The manuscript is well-written and presents the information in a clear and straightforward manner. I am concerned, though, that the new method for filtering DBS scans presented here is behind the state-of-the-art. For instance, Steinheuer 2022 (see below) introduces a method to filter DBS and VAD scans and adaptively reject low-quality measurements. An added benefit is that they use radial wind speed only, avoiding issues associated with fixing an SNR cut-off (which should vary with lidar settings). I would like to see more thought given to other methods such as the one they propose, or direct comparisons against results employing other methods. Or, if other methods are inappropriate for use on the lidar hardware itself given hardware limitations, that could be noted as well. That is, the proposed method might be preferred for running onboard the lidar given its simplicity. In any case, additional justification is needed for the current method compared to other methods.

J. Steinheuer, C. Detring, F. Beyrich, U. Löhnert, P. Friederichs, and S. Fiedler, "A new scanning scheme and flexible retrieval for mean winds and gusts from Doppler lidar measurements," Atmospheric Measurement Techniques, vol. 15, no. 10, pp. 3243–3260, May 2022, doi: 10.5194/amt-15-3243-2022.

*Answer:*

Thank you for sharing this article; it is indeed relevant in the context of data availability. However, performing a comparison with all the methods proposed in that paper is not feasible, as they used scanning lidars, which can measure many radial velocities on the cone base, as well as changing elevation and azimuth angles. In the BEAM 6x lidar, there are six different beams with fixed, preset elevation and azimuth angles, and the instrument does not provide any mechanical rotation or optical scanning of the beam direction.

To address the referee's comments, we have added a citation to the paper, as well as an explanation of the hardware limitation for comparing the adaptive method with other methods mentioned in that article.

*"Despite the variety of scanning patterns and suggested elevation and azimuth angles for wind lidars (Steinheuer et al., 2022), this study focuses on a fixed beam geometry due to the hardware limitation. Nevertheless, a similar algorithm could be applied to instruments operating on the same principle."*

Based on a simple comparison, we find that the adaptive method outperforms any fixed configuration (e.g., 6 or 3 beams, mentioned in Steinheuer et al., 2022) in terms of availability, because a fixed configuration represents only a subset of the possible combinations in the adaptive approach. This result is illustrated in Figure 6, which compares a fixed configuration of five beams (referred to as the standard method) with the adaptive method. Similar results and conclusions are expected in comparison with other fixed configurations. To address the referees' comment about comparison, the following sentence has been added to the manuscript:

*"A comparison with fixed configurations will logically show that wind speed availability will always be equal to or higher for the adaptive method, as a fixed configuration is merely a subset when the SNR constraint is satisfied."*

The study presented here has several aspects that reveal interesting insights and different points of view compared to Steinheuer et al. Below, we outline the main differences and provide corresponding details in the manuscript (written in double quotation marks below).

- **Validation at higher altitudes:**
  *"The work presented here offers several aspects compared to previous studies that reveal interesting insights and diverse perspectives. One aspect is that the validation of measurements is performed at higher altitudes, up to 244 m, where reduced availability makes accuracy assessment more demanding compared to lower altitudes (e.g., 90 m), where availability is nearly complete and accuracy is generally assured. Extending this validation to even higher altitudes would be especially valuable, as the differences between the accuracy of DBS methods are expected to become more pronounced as variations in availability are more significant."*

- **Challenges with Doppler fit residuals:**
  Although a fixed SNR threshold may not be ideal in all situations, it offers a simple and practical approach compared to more complex methods that require tuning multiple parameters. For example, in Steinheuer et al., one parameter depends on the residuals of the Doppler velocity fit, which can be challenging to interpret and tune, as it is difficult to distinguish between poor fits and turbulent flow.
  *"Another criterion that can be discussed is the residuals of the fit to Doppler velocities (Steinheuer et al., 2022), which can be calculated after performing DBS. We do not recommend relying on this metric, as high turbulence can also result in a high residual, making it difficult to differentiate between a poor fit and turbulent flow."*

- **Velocity retrieval despite partial beam blockage:**
  *"The adaptive method provides an additional advantage in terms of the beam blockage issue. If one beam is blocked by a solid object or otherwise compromised, velocities can still be*

*retrieved using a combination of the remaining beams. This addresses a common issue faced by clients, who sometimes must return complete datasets to lidar manufacturers for post-processing of the Doppler measurements when a single inclined beam is blocked or otherwise compromised, as no velocity reconstruction is possible in the current setup under these conditions. Using the adaptive algorithm, this problem is avoided, allowing the retrieval of velocity without the manufacturer's intervention, and also saving significant time and effort."*

All in all, we believe that the findings of this study are of interest to the wind lidar community and to wind farm developers, offering practical insights for measurement strategies and data processing.

---

## Author Comment (AC2)

**Author Comments:**

We would like to thank the referees for their valuable feedback, which has enhanced the clarity of this manuscript. We have carefully considered each comment and revised the manuscript accordingly. Our point-by-point responses are provided below.

**RC2: https://doi.org/10.5194/wes-2025-165-RC2, 30 Oct 2025**

**General comments:**

The authors provide a concise summary of a comparison of two 10-min average wind retrieval algorithms specific to the BEAM 6x lidar from Lumibird. The newly proposed retrieval includes individual DBS retrievals from scans where only a subset of the six beams contains data with backscatter signals with signal-to-noise ratios above a critical level at the relevant range gates corresponding to a wind retrieval height. The article seems directed to justify the implementation of this algorithm for a commercial product, and some additional information could be included to make the article academically more relevant.

**Specific comments:**

The algorithm is specifically designed for 10-min mean winds. These average retrievals will ultimately include both, individual retrievals from a full, successfully retrieved set of six radial velocities, and a subset of now included DBS retrievals based on fewer radial velocities. While the data shown (10-min averages) may suffice to market a machine using this updated algorithm, a reader may also be interested in a comparison of the retrievals from sets of a) the full six radial velocities, b) 5 valid velocities, c) 4 , etc. Adding this will allow the reader to evaluate the influence of these added members of the average, and the data shown in Fig 7 becomes more valuable from a more academic standpoint. Similarly, there may have been results published of experiments where subsets radial velocities were omitted from DBS and/or VAD retrievals. If so, these should be cited in this context, and differences in results should be discussed.

*Answer:*

The adaptive algorithm is specifically designed to enhance **instantaneous** availability (please refer to Equation 3). Figure 6 illustrates the instantaneous availability achieved by the adaptive method compared to the standard method (which uses only a fixed configuration of beam numbers 1 to 5). However, the accuracy for both wind speed and wind direction is presented over 10-minute average intervals, because it is commonly referred to in IEC standards and calibration reports.

Following a comparison, we find that the adaptive method logically outperforms any fixed configuration (e.g., 6, 5, 4, or 3 beams) in terms of availability, since those fixed configurations are always subsets of the adaptive approach. This conclusion is also evident in Figure 6, where the adaptive method applied to the real measurement is compared with a fixed configuration of five beams (standard method).

To address the referee's comment and improve clarity, we have added the following sentences to the manuscript:

*"In this algorithm, the objective is to increase instantaneous availability. A comparison with fixed configurations will logically show that wind speed availability will always be equal to or higher for the adaptive method, as a fixed configuration is merely a subset when the SNR constraint is satisfied."*

Steinheuer et al. (2022) presented several scanning patterns and compared the availability of VAD and DBS for recommended elevation and azimuth angles. We have added a citation to their work and briefly discussed the relevant differences and limitations. Please see also our response to the first referee.

The description of the BEAM 6x lidar (Line100ff) should be moved to an earlier section as this information is needed prior to the "Methodology" section as the proposed retrieval is so some extend specific to that (or similar) lidar model. For example, you mention 10-min averages on line 71, but the reader does not know how many individual retrievals are contained in such an average until this information is given (i.e. 5.5-s scan repetition).

*Answer:*

Thank you for this recommendation. This paragraph has been moved to the earlier section, preceding the description of the adaptive method.

There are several aspects that could be clarified: The "sliding window" (Line 53), does it slide across the 2.6 m range gates of each of the six beams, or does this mean that winds are retrieved at vertical increments (i.e. heights) at these intervals?

*Answer:*

Thank you for your comment. In this context, the term "window" refers to the set of six radial velocity measurements taken sequentially in time, one from each of the six beams, at a given height. The sentence has been rephrased as follows to improve clarity:

*"The algorithm begins by applying a sliding time window to Doppler velocities. At each height, the window contains six consecutive Doppler velocity measurements, one from each of the six beams."*

It is stated that the addition of lidars increases the wind retrieval height (i.e. to 1000 m; line 97), but the range gates are listed as 2.6 m. How does adding lidars lead to extended ranges? Are they run with different gate lengths, and if so, which are chosen?

*Answer:*

All three lidar units provided separate measurements, and there was no combination or fusion of their measurements. This option is available in the software setup to select a maximum measurement range that exceeds the design range (500 m). The following sentence is added to explain this extension:

*"All three lidar units operated independently, and the BEAM 6x system allows users to specify a maximum range that exceeds its 500-m design range."*

Several formulations could be more precise to adhere to scientific journal standards. For example, on line 27, you state: "doppler velocities"… fall "below"…"SNR threshold". Isn't it the backscatter that is below the SNR? See also line 51 states a bit colloquially:"beams fall below SNR". In the paragraph starting with line 49, it is omitted that the described method is applied to each range gate. I'd like to encourage the authors to scan the manuscript for similar, slightly imprecise, formulations, and to improve them.

*Answer:*

Thank you for pointing out these imprecise formulations. We revised the manuscript to clarify that the quality of backscattered signals, not Doppler velocities or beams, drops and falls below the SNR threshold. Also, we specified that the method is applied to each range gate. All corrected sentences are highlighted in the track-changed version.

**Technical corrections:**

Line 124: add space:"analgorithm" -> "an algorithm"

*Answer:*

Thanks for pointing out this typo. It has been corrected accordingly.